# A Machine Vision Method for Identifying Blade Tip Clearance in Wind Turbines

**DOI:** 10.3390/s24185935

**Published:** 2024-09-13

**Authors:** Le Zhang, Jiadan Wei

**Affiliations:** 1Wuxi Key Laboratory of Intelligent Robot and Special Equipment Technology, Wuxi Taihu University, Wuxi 214064, China; 2College of Automation Engineering, Nanjing University of Aeronautics and Astronautics, Nanjing 211106, China; weijiadan@nuaa.edu.cn

**Keywords:** machine vision, blade tip clearance, wind turbine, real-time trajectory, fast Fourier, transform analysis

## Abstract

This paper introduces a machine vision method for measuring the blade tip clearance in a wind turbine. An industrial personal computer (IPC) is installed in the nacelle of the wind turbine to continuously receive video data from a digital camera mounted at the bottom of the nacelle. Using the open-source computer vision (OpenCV) digital image processing library data base, the real-time trajectory of the turbine blades is determined from the video data. Furthermore, fast Fourier transform (FFT) analysis is performed for determining the operating frequency of the blades in the images. The amplitude analysis performed at this operating frequency reveals the pixel-based blade tip clearance, which is then used to calculate the actual clearance of the wind turbine. This value is subsequently transmitted to the main controller of the wind turbine. The main controller can enhance the operational safety of the wind turbine by implementing appropriate pitch control strategies to restrict and safeguard the blade tip clearance. The results obtained by conducting experiments on a 2.0 MW wind turbine unit validate the effectiveness of the proposed identification method. In this method, the blade tip clearance can be calculated effectively in real time, and both the video sampling rate and communication speed meet the requirements for controlling the blade pitch.

## 1. Introduction

In the renewable energy industry, the primary objective across the entire value chain is to reduce the per unit cost of electricity generation, known as the cost of energy (CoE). When the CoE of renewable energy falls below that of fossil fuels, it becomes possible to avoid our reliance on the latter and subsidies for the former, thus paving the way for green and sustainable electricity generation. In the case of wind energy, the most effective method for reducing CoE is to use longer blades and taller towers. Wind farms worldwide are increasingly established in regions having medium to low wind speed. This necessitates an increase in the blade length to capture more wind, but this also leads to an increase in the mass of the blades. The design envelopes require the deflection of the fully loaded blades to not exceed the specified clearance limit between the blade tip and tower. According to the DNV-GL standard requirement [1], the blade tip clearance during wind turbine operation cannot exceed 30% of the clearance under no-load conditions. However, the current sensor technology and clearance-controlling techniques employed in the blades of wind turbines lack the capability to measure clearance reliably [2]. If a technology can be developed to measure the blade tip clearance of an operating turbine at a low cost, then larger blades can be designed and manufactured, leading to a reduction in the CoE.

For monitoring blade health, Refs. [3,4] applied the filtering optimization algorithm to identify various defects on blades, such as scratch-type, crack-type, sand-hole-type, and spot-type defects. However, these methods cannot measure the blade tip clearance directly. Since fiber optic sensors are insensitive to electromagnetic interference and immune to lightning strikes, monitoring methods using fiber Bragg grating (FBG) sensors are applied to wind turbine blades. For instance, FBG sensors are used to detect blade damage or monitor blade structural responses, such as bending loads. In some works [5,6,7,8,9,10], stress and frequency information from blades under both static and dynamic loads were sampled by affixing FBG strain sensors on the surface of the wind turbine blades. In order to ensure a comprehensive blade assessment, the number and length of FBG sensors need to be increased proportionally with the blade’s length, typically one sensor per meter along the entire blade length. Choi et al. [11] compensated temperature variations in the measured values of the FBG sensors to achieve more precise measurements of blade deformation. Pressure strain gauges displayed similar high-precision characteristics to fiber optic sensors in measuring object deformation at a lower cost. In Refs. [12,13], strain gauges were applied on the blade surface, and a simplified practical model was established based on a linear relationship between the blade deformation and deflection to facilitate the detection of blade deformation during operation. Micro-electro-mechanical system (MEMS) devices such as accelerometers and gyroscopes have also been used in prototype setups only due to their vulnerability to lightning strikes and low feasibility of maintenance (requiring more installed in the various positions inside the blades). In Ref. [14], MEMS sensors were embedded within blades to measure their rotation angles, and also to predict their tip clearance by using aerodynamic and neural network modeling approaches. These methods can simulate and compute the blade tip clearance by establishing a comparative relationship between the blade deformation and aerodynamic models. Due to the difficulty in installing sensors internally or externally on blade surfaces, these methods are currently applied in the prototype development stages only. In Ref. [15], a laser displacement sensor was installed in a tower to monitor the distance and angle of the blades each time they passed through it. High accuracy is the advantage of this approach. However, when blades rotate continuously with the yaw control of the turbine, it becomes challenging for the laser scan approach to detect the blade tip clearance at all the yaw angles. For quality inspection during production, Anderw et al. [16] obtained surface deformation profiles of blades by conducting high-speed scanning with a coherent laser radar. Despite having high measurement accuracy, the high cost and bulky size of the coherent laser radar prevent its practical application in measuring blade tip clearance. In Ref. [17], blade deformation was monitored by using one ground-based camera sensor, and then, the blade-to-tower clearance was calculated by comparing the monitored data with aerodynamic simulation data of blades. However, the studies did not address the synchronization issue between the blade position and rotation of the nacelle. Moreover, due to the ground-based installation of cameras and communication delay in data transmission to the controller of the wind turbine, it is impossible to provide real-time clearance data to the controller. As a result, the effective and timely control and protection of blade tip clearance cannot be ensured. In Refs. [18,19,20], natural frequencies and deformation frequencies were extracted by converting the temporal frequency of displacement to monitor the blade health, such as the detection of missing root bolts. A new concept based on ultra-wideband (UWB) radio was proposed in [21]. By installing a pair of UWB sensors at the root and tip of the blades, the time difference between sensor transmission and reception signals was calculated in real time, subsequently allowing the computation of the position of the blade deformation. Although it has been tested and proven to be correct, many practical issues still need to be addressed during its application. Table 1 shows the comparison of the above-mentioned methods used to calculate blade tip clearance.

From the above survey, it is clear that there is a need for a cost-effective system for measuring blade tip clearance, which can operate in all weather conditions, requires minimal daily maintenance, and gives clearance in real time, while mitigating the problem of low transmission delay to the controller of the wind turbine.

In this paper, a method based on machine vision is proposed for measuring the blade tip clearance in wind turbines. In this method, a digital camera is installed at the bottom of the nacelle to observe the positions of the tower and blades, and an IPC is placed inside the nacelle to continuously read the video data transmitted by the digital camera. Using machine vision algorithms, the IPC determines the trajectory of the blades and then calculates the blade-to-tower clearance. The remainder of this paper is organized as follows: The working principle of machine vision in measuring blade tip clearance is explained in Section 2. A method for the offline calibration of parameter measurement systems is presented in Section 3. In Section 4,OpenCV is employed to develop a blade dynamic recognition algorithm and carry out FFT analysis of blade clearance, and also to perform data analysis during both daytime and night-time. The field experiments, validation, and data analysis are presented in Section 5. Finally, in Section 6,the present work is concluded along with outlines of some future research directions being presented.

## 2. Operating Principle

Machine vision is a rapidly developing branch of artificial intelligence. It has found successful applications in many areas, such as autonomous driving, intelligent transportation, automatic logistics sorting, and smart security. It plays a crucial role in many industries, such as manufacturing, transportation, and security, among others. Machine vision is a comprehensive technology that encompasses digital image processing, mechanical engineering techniques, control systems, optical imaging, analog and digital video technology, and computer hardware and software. The architecture of machine vision-based systems primarily consists of hardware components and software algorithms. The hardware components include a light source system, lens, camera, image acquisition card, and vision processor. The software packages mainly involve some traditional digital image processing algorithms and deep learning-based image processing algorithms.

The principle of measuring the blade tip clearance using machine vision relies primarily on a webcam that captures the relative positions of the blades and tower in each frame. Moreover, an image recognition algorithm is applied to extract the trajectory of each blade as it passes the tower. After that, the FFT algorithm is employed to extract the frequency characteristics of the blade trajectory by automatically identifying the trajectory coordinates of the blade tip in the images. These coordinates are then compared with the relative coordinates of the tower, yielding the pixel-based clearance within the images. Subsequently, the ratio between the pixel-based clearance and the physical spatial relationship is considered to solve the actual physical clearance between the blades and tower.

The blades of an operational wind turbine usually rotate in a clockwise direction, as shown in Figure 1a, where the three blades periodically pass through the tower. The minimum distance from the blade tip to the tower, known as the blade tip clearance, is measured when the blade comes in the vertically downward position and becomes parallel to the tower, as illustrated in Figure 1b by ∆Z. Facing vertically downward, the camera is mounted at the bottom of the nacelle positioned between the tower and hub. The dashed area represents the field of view of the camera, which includes the regions of both blades and tower. The camera is mounted on the central axis of the nacelle, as shown in Figure 1a, ensuring that the tower remains in the center of the screen during the video recording. Whenever a blade passes through detection area of the camera, it captures clear images covering the distance between the blade and tower. This information is transmitted in real time via communication cables to the IPC located inside the nacelle, which uses a built-in image recognition algorithm to calculate the clearance in the video and then transmits it to the main programmable logic controller (PLC) of the wind turbine to control and protect the blade tip clearance.

## 3. Calibration Angle and Image-to-Physical Space Ratio Parameter Identification Offline

In Figure 2, the left image represents the relationship between the video/images captured by the camera and real-world objects. The basic unit of image size is a pixel, which is composed of two parameters: width and height. For example, an image with a resolution of 640×480 means that the image consists of 640 rows and 480 columns, totaling 307,200 pixels in a pixel matrix. The real-world dimensions corresponding to the image are measured in meters. Once the real dimensions of an object in Plan 2, such as the wind turbine’s tower base diameter (D) and the pixel sizes on an image frame, are determined, the proportionality coefficient A_1_ between them in meters per pixel can be established. On the right side of Figure 2, the detection range of the camera is depicted with dashed lines. The camera is mounted at the bottom of the nacelle at a vertical height of H_1_ above the ground. Two-dimensional images within the dashed lines are recorded on two planes, namely Plane 1 and Plane 2, relevant to the clearance calculation. Plane 1 represents the plane at the minimum vertical height of H_2_ of the blade tip above the ground. The minimum distance ∆Z between the vertical projections of the blade and tower on Plane 1 is the blade tip clearance. Plane 2 corresponds to the horizontal position at the ground level. Based on the trigonometry depicted in Figure 2, the proportional relationship A_2_ between the real size of the object on Plane 1 and the pixel size on the image can be calculated as expressed by Equation (1).
(1)$$A_{2}=\frac{\left(H_{1}-H_{2}\right)}{H_{1}}\cdot A_{1}$$

In this context, Plane 1 represents the foreground, where only the dynamic trajectory of the blades as well as the information about the stationary tower needs to be captured. On the other hand, Plane 2 is the background that primarily consists of ground-level objects, including the tower, tower base, and planted trees and vegetation. Objects in the background either remain nearly static in the video/images or exhibit minimal displacement, such as slight swaying of trees due to the wind. Given a significant difference in the positional changes between the foreground and background objects in video/images, an algorithm for motion detection from image recognition can be employed to distinguish the dynamic blade motion, while appropriate blurring and filtering methods can be used to eliminate any interference noise caused due to the slight movement of the background objects.

The block diagram of the proposed method is shown in Figure 3. There are three primary steps. The first step focuses on obtaining raw data, including the frames per second(FPS) of video files, the pixel size of each frame file, the installation location of camera, blade heights (H_1_, H_2_), the base diameter of the tower(D), and raw data recorded by the camera. In the second step, offline image processing is performed to calculate calibration parameters, such as A_1_, A_2_, the tilting angle of the camera (β) with respect to the nacelle axis, and the reference point (P1) of the tower base in the image frame. These calibration parameters are stored in the corresponding registers of the IPC to make real-time corrections in calculations. Subsequently, the online object motions algorithm is applied by the IPC to identify the blade trajectories and extract the frequency and amplitude characteristics of the blade motion using the FFT algorithm. By analyzing the motion frequency and amplitude variation threshold of the blades, the clearance ∆z in the image is concluded and then converted into the actual physical clearance ∆Z. In the third step, the main control system of the wind turbine, which is a programmable logic controller (PLC), receives the calculated real clearance from the IPC. These clearances are used to control and protect the blade clearance in the wind turbine, so as to ensure its safe operation. These three steps are carried out independently in the camera, IPC, and PLC devices, respectively, to process and calculate relevant information. In addition, the data transmission is facilitated through a communication bus.

It is necessary to calibrate the parameters related to the installation of the camera and the tower of the wind turbine. This can be achieved by obtaining A_1_, A_2_, camera-to-nacelle axis tilt angle β, and reference point P1 of the tower base in the image frame through offline image processing.

The calibration process is illustrated in Figure 4 as follows:

It is assumed that the video recording rate of the camera is 30 FPS.

S1: Capture video streams from the camera, and extract multiple images containing at least three consecutive images of the wind turbine blades.

S2.1: Extract the blades from the images and combine them into a single image to serve as a calibration image.

S2.2: Connect the trajectories of the blade tips, and measure angle β as the camera installation tilt angle between the trajectory line and bottom edge of the image frame.

S2.3: Find the number of pixels representing the tower base diameter D in the image, and calculate the ratio between its actual value and the number of occupied pixels. This ratio is defined as the scaling factor A_1_, measured in meters per pixel.

S2.4: Define the coordinate of the origin at the top-left corner of the image, establish a coordinate system (X0, Y0), and record the central position of the tower base as point P1.

S2.5: Locate the tower reference surface at height H_2_ by drawing a horizontal line parallel to the blade tip trajectory, which will serve as the tower reference. The distance between this reference and the blade tip trajectory is denoted by ∆z.

S3.1: Rotate the image coordinates (X0, Y0) counterclockwise by an angle β with respect to the central position of the tower base, thereby transforming the (X0, Y0) coordinate system into the (X1, Y1) coordinate system, where the trajectory of the blades becomes parallel with the bottom plane, as expressed by Equation (2).
(2)$$P_{x1y1}=P_{x0y0}\;\left[\begin{array}{cc} cos\left(\beta \right) & \sin\left(\beta \right)\\ -sin\left(\beta \right) & cos\left(\beta \right) \end{array}\right]$$

S3.2: Further, optimize the (X1, Y1) coordinate system into the (X2, Y2) coordinate system as expressed by Equation (3). This optimization compresses the pixels of the image along the Y-axis based on its distance from the top edge of the image to the tower reference surface. This reduces the space required for storing the data of the image and also optimizes the calculations to be performed by the processor.
(3)$$P_{x2y2}=P_{x1y1}\;\left[\begin{array}{cc} 1 & 0\\ 0 & 1+y0/P_{x1y1} \end{array}\right]$$
where y0 is the pixel distance from the top edge of the image to the tower reference surface in the (X1, Y1) coordinate system.

S3.3: Based on the installation height of the camera (H_1_), the height of the blade tip above the ground (H_2_), and proportionality coefficient A_1_, calculate the proportionality coefficient (A_2_) for the actual distance in the H_2_ height plane in relation to the pixel size on the image.

S3.4: Store A_2_, β, P1, and y0 in the registers of the IPC for the subsequent real-time calculation of clearance ΔZ.

## 4. Algorithm for Online Recognition of Blade Motion

Due to the continuous nature of video sequences captured by the camera, consecutive frames will exhibit minimal changes if there is no moving object in the captured scene. However, in the presence of moving objects, such as rapidly rotating wind turbine blades, there are noticeable variations between two consecutive frames. Differencing two consecutive frames pixel-wise and taking the absolute value of their grayscale difference, it is possible to detect moving objects if the difference exceeds a predefined threshold. These detected movements are then saved as motion images. OpenCV is a freely available software library used extensively in computer vision and machine learning applications. The OpenCV branch identifies the challenges that require resolution for capturing an image and ascertaining the presence of motion. OpenCV was originally developed as an Android application subsystem for capturing, processing, and analyzing images [22], which has now evolved into a comprehensive open-source library for real-time computer vision, machine learning, and image processing [23]. It provides interfaces for popular programming languages, such as Python, Ruby, and MATLAB, thus making it versatile for implementing different image processing and computer vision algorithms. The functions of OpenCV can be utilized for performing mathematical operations in identifying blade trajectories in image frames.

In order to check whether any motion is present in the live images captured by the webcam, they are compared with each other. In this way, the changes in those frames can be detected, making it possible to forecast the occurrence of any motion. A flowchart illustrating this process is shown in Figure 5.

Images are extracted from the video stream acquired by the camera and saved in the IPC at a fixed interval of 1 s, as depicted in Figure 6a. All these image packets have a pixel resolution of 1024 × 576 and provide clear visual information about the wind turbine’s tower, blades, tower reflection in sunlight, and ground vegetation. Employing the frame difference method, as illustrated in Figure 5, the blade movement between consecutive frames in an image packet is recognized, as shown in Figure 6b, where the highlighted areas represent the trajectories of the blades. Since the blades usually rotate clockwise, their movement trajectory in the image is seen from the right side to the left side. The darker background indicates that the information about the wind turbine’s relatively stationary tower, tower reflection, and ground vegetation is filtered out.

Next, the camera’s tilting angle β and the coordinates of the tower reference surface and base center P1 are extracted from the registers of the IPC. Then, the image is rotated counterclockwise through angle β about the base center P1. Subsequently, the image is cropped and compressed by adjusting the position of its top edge to coincide with the tower reference surface. This cropping involves the trimming of a length of y0 pixels from the top edge along the Y-axis. In addition, a portion of the pixels from the bottom edge is also removed by retaining only the pixels up to the blade root. This compression aims to reduce data size without affecting the calculation of blade tip clearance ∆z. After the rotation and cropping of coordinates, the pixel resolution of the image becomes 1024 × 205, as shown in Figure 6c.

Figure 6c allows human observers to clearly discern the motion trajectory of blades by drawing auxiliary lines. It is also possible to measure the blade’s tip clearance ∆z. But the IPC still cannot automatically recognize the coordinates of the blade tip in the image. Although it can be seen in Figure 6c that the motion trajectory of the blades contains distinct and regular frequency information, its background noise exhibits irregular white noise-like spectra. The image can be analyzed row by row using FFT to read the frequency information contained by it and also to locate the inflection points in frequency changes, which correspond to the position of the blade tip on the Y-axis. Figure 6d shows the result of this row-by-row FFT analysis. Since it is performed row by row, the horizontal axis (X-axis) represents frequency, the vertical axis (Y-axis) represents pixel positions in the image, and the grayscale values of the pixels in the image indicate the amplitudes of different pixels along the Y-axis corresponding to different frequencies. The frequency analysis reveals that there are almost no data below 8 Hz in the image. At 16 Hz, the pixel point of the highest brightness appears. Significant bright spots can be seen at multiples of 16 Hz, such as at 32 Hz and 48 Hz, indicating that the composite frequency of the blade’s motion trajectory is around 16 Hz.

Then, the grayscale (brightness intensity) curve at 16 Hz is plotted, as shown in Figure 6e. Below a point on the Y-axis having the coordinate of 55, the grayscale value is nearly zero. Starting from that point, the grayscale increases sharply, reaching its maximum value of around 9. Hence, the threshold value for the inflection point on change in grayscale is set to 1. Any point on the Y-axis exceeding this threshold value is considered as the position of the blade tip. This point on the Y-axis corresponds to the pixel-based clearance ∆z. After that, parameter A_2_ is read from the registers of the IPC. Hence, ∆z*A_2_ represents the actual clearance ∆Z in the physical space.

This methodology enables automatic image recognition and calculation of the tip clearance of blades. Once the measured clearance ∆Z is received from the IPC, the wind turbine controller (PLC)can control the tip clearance of the blade by adjusting its pitch angle [24]. It ensures the absolute safety of the blade tip clearance by mitigating the risk of tower collisions and complying with relevant design standards.

In order to accommodate those scenarios where blade data need to be collected at night, supplementary lighting is added near the camera at the lower section of the nacelle. The direction of the supplemental light matches with the measurement direction of the camera and serves to augment the light intensity during night-time or in low-light conditions. Figure 7 illustrates the analysis and processing of the night-time clearance measurement with the supplemental lighting. The image processing and blade recognition methods remain consistent with those used during the daytime measurement as shown in Figure 6. A comparison between the night-time and daytime data reveals several characteristics which are as follows:(1)Due to the limited coverage of the supplementary lighting, it illuminates only the vicinity of the tower’s center area. Consequently, the camera can recognize only the blades within the illuminated area. Within the same duration of operation, the reduced detection area results in a significantly lower number of blades being recognized. In the detection area shown in Figure 7, only four blades can be identified clearly.(2)The quality of blade recognition is higher in the vicinity of the supplemental lighting center, and it becomes blurred towards the edges of the illuminated area. Typically, in a data packet of a 1 s video, only three to four complete blades can be recognized. In principle, a minimum of two accurately recognized blades are required to fit the motion trajectory of blades and calculate the blade tip clearance. The intensity of the supplemental lighting and minimum detection angle can be adjusted based on the rotational frequency of the blades.(3)It is found in the FFT analysis of the blades that the FFT amplitude is lower during the night-time, as seen in Figure 8, compared to that during the daytime under good lighting conditions, as seen in Figure 6. This indicates a lower confidence level in image recognition during the night-time.(4)The frequency inflection points in the FFT analysis remain distinct and steep during the night-time, allowing the algorithm to accurately calculate the clearance between the blades and tower.

## 5. Experimental Validation and Data Analysis

Using the above-stated method, experiments were conducted to measure the blade tip clearance in a newly installed 2 MW wind turbine generator. Table 2 presents the main specifications of the tested wind turbine and camera sensor.

For offline calibration, the required parameters of the wind turbine were set as follows: H_1_ = 76 m, H_2_ = 21 m, β = 7.6°, A_1_ = 0.1346 m/pixel, A_2_ = 0.0974 m/pixel, y_0_ = 103 pixels, and coordinates of P1 = (483, −232).

Figure 8 shows the relationship between the measured clearance and wind speed. Over a duration of 48s, as displayed in the chart, the blades of the wind turbine pass 25 times through the detection area of the camera, with an average of one blade passing through the detection area every 2 s. The processing of the camera data and calculation of the blade clearance in the IPC is performed every 1 s. Consequently, during the detection of 25 blades, clearances are measured in the range from 6m to 8m corresponding to the wind speeds in the range from 8 m/s to 12 m/s. Without considering the factors related to the variation in the pitch angle of the blades, their tip clearance is found almost inversely proportional to the wind speed. As the wind speed increases, the clearance of the blade tip decreases.

Figure 9 shows the comparison of the measured clearances with wind speeds in a 24 h period, during which the wind speeds vary from 2.5 m/s to 12 m/s, and the clearances vary approximately from 12 m to 5 m. Throughout the entire range of the wind speed, the clearance could be measured accurately. In a low range of wind speed (2–8 m/s), the blade tip clearance is inversely proportional to wind speed, ranging approximately from 6.5 m to 11 m. As the wind speed increases, the blade tip clearance gradually decreases to 6 m. At a wind speed of 9 m/s corresponding to the turbine’s rated full load and maximum blade pitch angle, the blade pitch angle decreases with increasing wind speed, leading to a reduction in the blade tip clearance.

To verify the measurement accuracy of the proposed scheme, a fixed laser displacement radar was installed on the experimental wind turbine to directly measure the clearance distance of the blades (a method similar to that described in [15]). Considering the high measurement accuracy of the laser displacement radar and the direct alignment of the radar’s installation position on the tower with the blade tip, the measurement data from the laser displacement radar can be considered as the actual clearance distance. Figure 10 shows the installation position of the laser displacement radar on the tower surface, and the radar’s measurement signal was transmitted to the main control PLC via a communication bus to be compared with the measurement data from the proposed scheme.

As shown in Figure 11, the imaging method is marked by an orange color, and the results obtained from the laser displacement radar are marked by a blue color. The trends in the results measured by the two methods were completely consistent, but it can also be observed that there were small discrepancies between the two measurement methods when the clearance distance was at a close range of 6–8 m. As shown in Figure 12, the comparative error defined as the value of laser measurement minus camera measurement falls between −0.7 m and 0.4 m. The cause of the error may be due to the calibration error of the tower center point in the image analysis method. Typically, the clearance protection distance for blades is 3.5 m (approximately30% of the clearance under no-load conditions). Therefore, the proposed algorithm meets the precision required to control the blade protection.

Through the testing verification, the proposed scheme has the following advantages over the traditional scheme, which can meet the application of blade clearance monitoring and protection:(1)Simple installation position and low maintenance cost.(2)Mature sensor technology at a lower cost.(3)Data can be detected and acquired within a 360° yaw angle.(4)The algorithm can meet the requirements of real-time clearance detection, thus supporting data required for controlling the blade protection.

## 6. Conclusions

This paper presents several innovations and contributions, as follows:(1)Blade tip clearance in a wind turbine is measured in real time using an industrial-grade digital camera and an IPC.(2)Transmitted camera images are dynamically tracked using the OpenCV digital image processing library for the identification of motion trajectories of the blades.(3)Frequency and amplitude distribution of the blade motion frequencies are determined pixel-wise by performing FFT analysis row-by-row. The amplitudes of the pixel column corresponding to the blade motion frequencies are extracted, and the clearances in pixels are calculated, yielding the actual clearances.(4)A cost-effective approach is employed to address the challenge in the measurement of blade clearance, thus laying the foundation for the protection of the blade tip clearance and wind turbine blade and tower load optimization, as well as cost optimization.

Attempts may be made in the future to address the challenges related to extreme weather conditions, such as heavy snow, fog, and rain. In such conditions, the collected images may suffer from low quality and may also contain dynamic objects, like snowflakes or raindrops. Research efforts are necessary to improve the filtering algorithms and minimize the errors in measuring clearance.

## Figures and Tables

**Figure 1 sensors-24-05935-f001:**
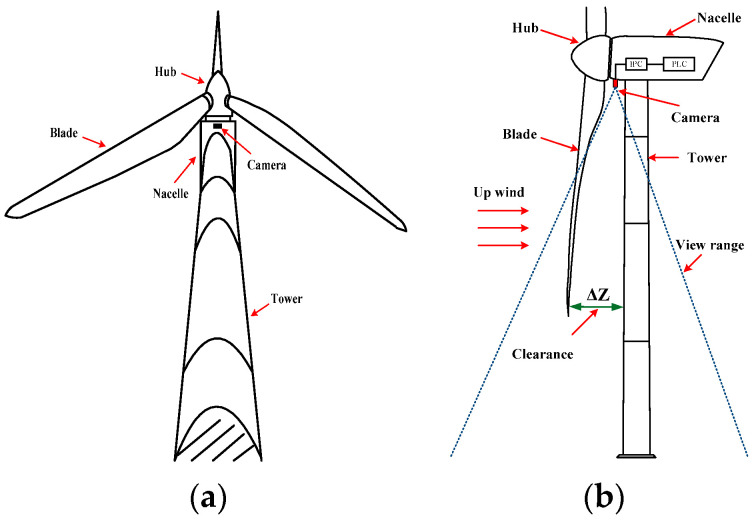
Schematic of camera installed in wind turbine.(**a**) Bottom view. (**b**) Right view.

**Figure 2 sensors-24-05935-f002:**
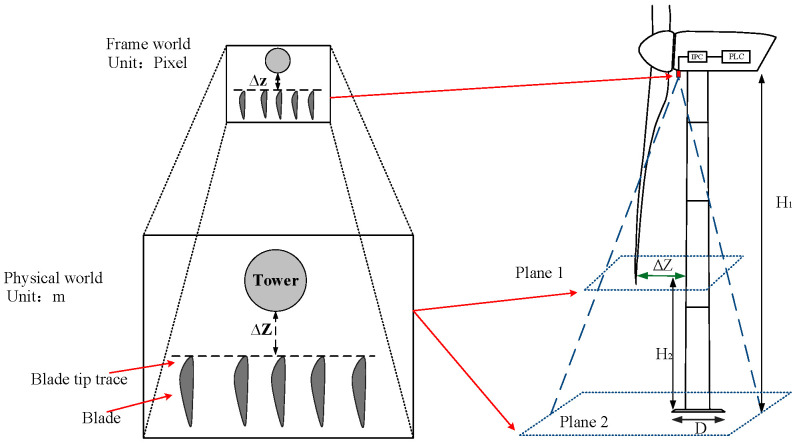
Schematic diagram of detecting range of installed camera and video imaging (top view).

**Figure 3 sensors-24-05935-f003:**
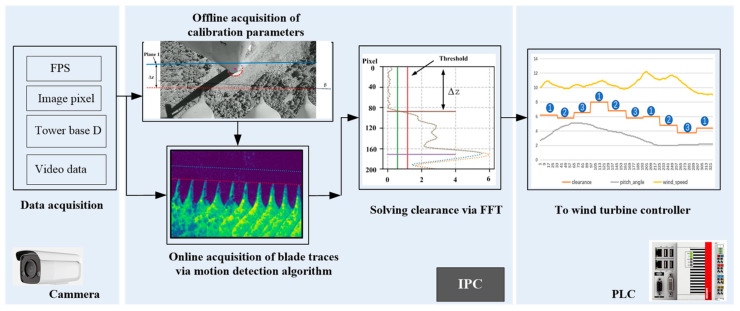
Block diagram of blade clearance acquisition based on image processing.

**Figure 4 sensors-24-05935-f004:**
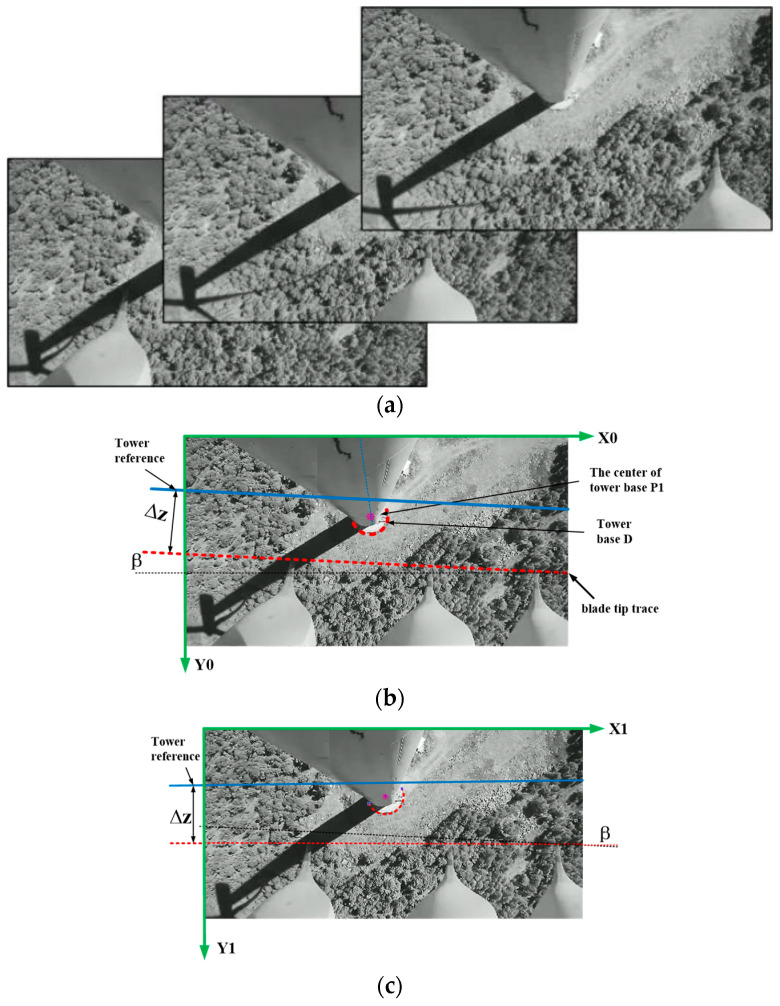
Schematic diagram of scaling and coordinate rotation.(**a**) Three adjacent images during transformation. (**b**) Image synthesis and reference point selection. (**c**) Image rotation, cropping, and compression.

**Figure 5 sensors-24-05935-f005:**
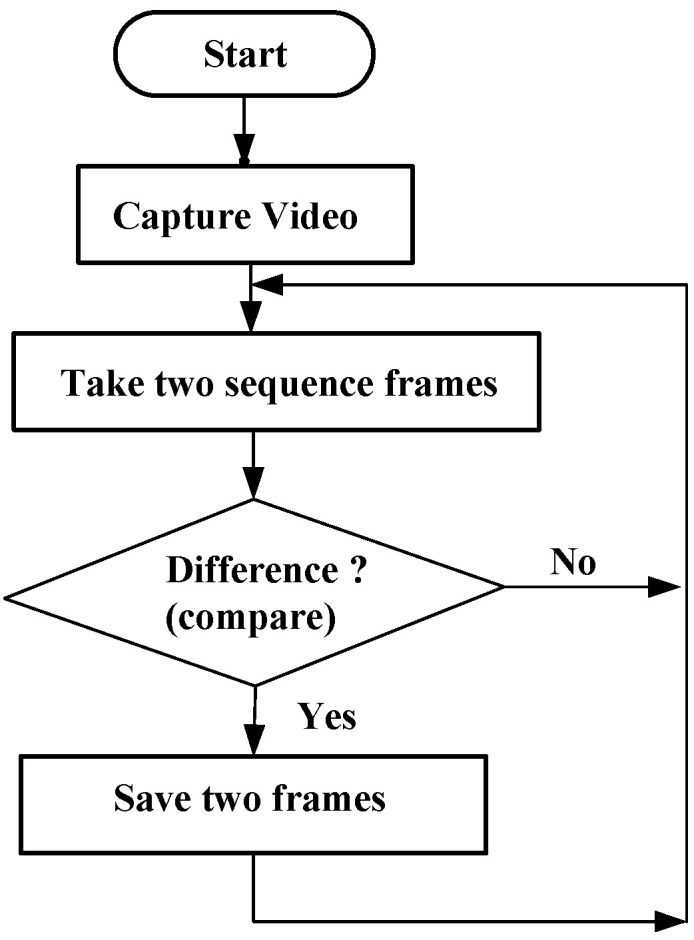
Flowchart of online calculation of blade motions.

**Figure 6 sensors-24-05935-f006:**
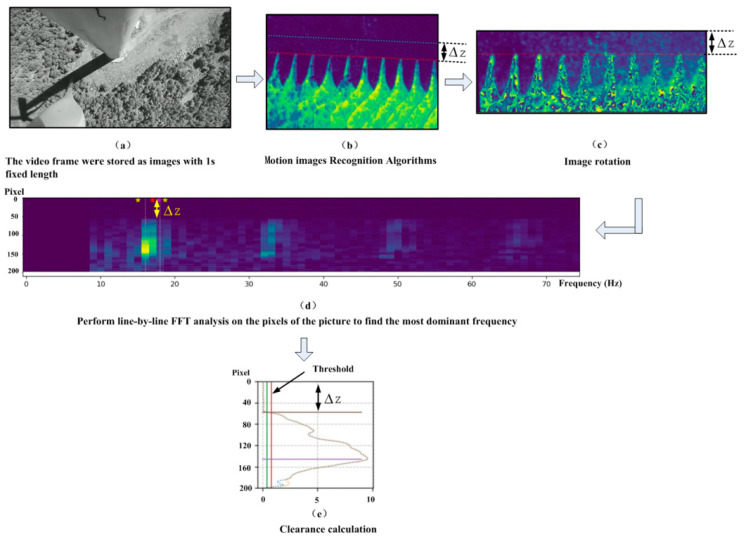
Schematic diagram of clearance calculation.

**Figure 7 sensors-24-05935-f007:**
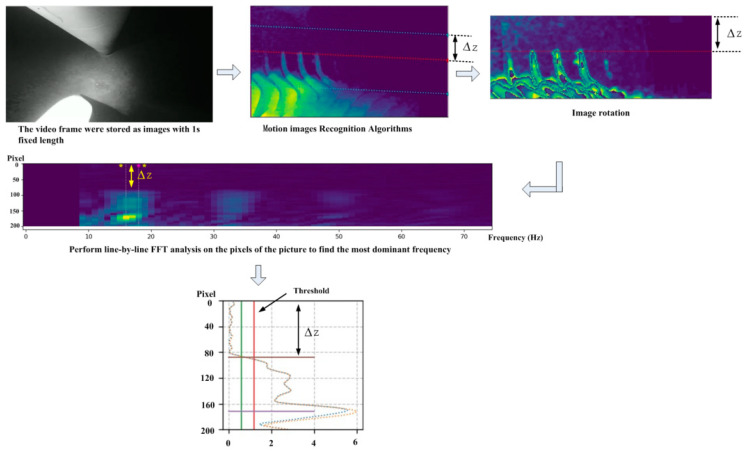
Schematic diagram of night-time image processing.

**Figure 8 sensors-24-05935-f008:**
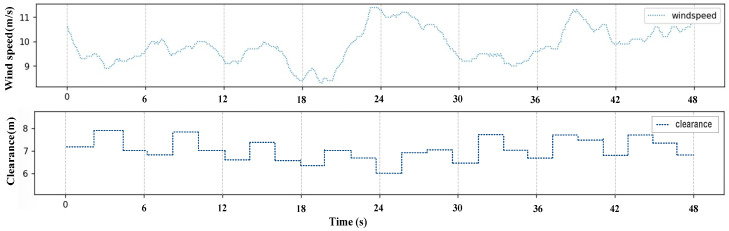
Relationship between measured clearance and wind speed.

**Figure 9 sensors-24-05935-f009:**
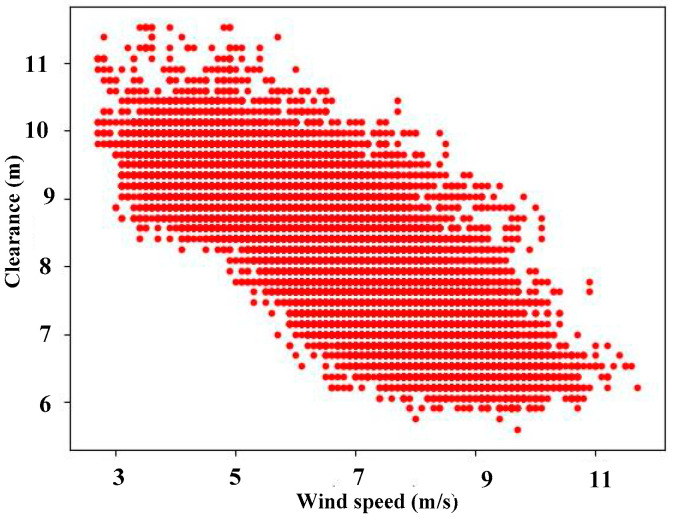
Comparison of the measured clearances with wind speeds in a 24 h period.

**Figure 10 sensors-24-05935-f010:**
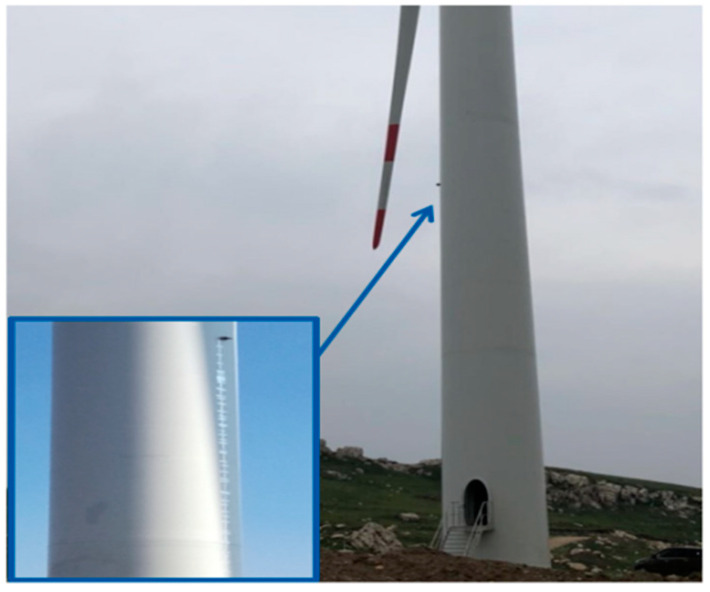
Installation position of a laser displacement radar.

**Figure 11 sensors-24-05935-f011:**
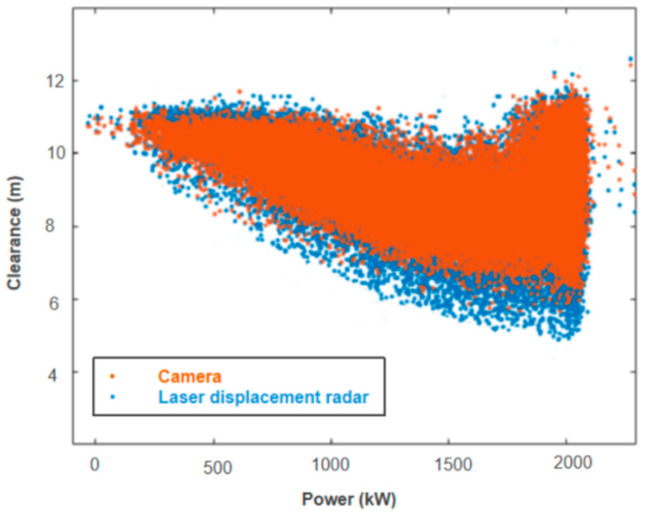
Clearance distances measured by the two methods.

**Figure 12 sensors-24-05935-f012:**
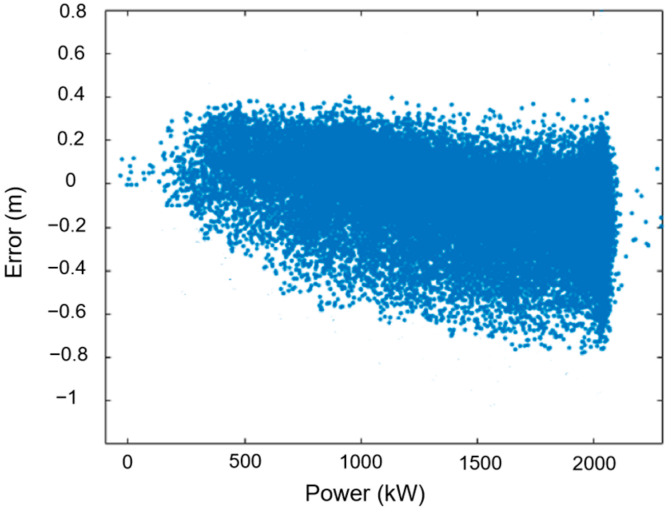
Differences between the results of the two measurement methods.

**Table 1 sensors-24-05935-t001:** Comparison of various methods.

References	Sensor	Install Location	Cost /Maintenance	Accuracy	Realtime Monitor	Status
[3,4]	Camera	Unmanned drone	High/difficult	Medium	No	Prototype development
[5,6,7,8,9,10,11]	FBG	Blade’s internal or external surface	High/difficult	High	yes	Prototype development
[12,13]	Strain gauges	Blade’s internal or external surface	High/difficult	High	yes	Prototype development
[14]	MEMS	Blade’s internal or external surface	High/difficult	High	yes	Prototype development
[15,16]	Laser radar	Tower’s external surface	Medium/Easy	High	yes	Functional validation
[17,18,19,20]	Camera	On ground	Low/Easy	Medium	No	Prototype development
[21]	UWB	Blade’s internal or external surface	High/difficult	High	yes	Functional validation

**Table 2 sensors-24-05935-t002:** Parameters of wind turbine and installed webcam.

Parameters	Value
Wind turbine rated power	2000 kW
Hub height	76 m
Rotor diameter	110 m
Blade rotation speed	10.2 rpm
Tower base radium	7.4 m
Cut-in wind speed	2.3 m/s
Cut-out wind speed	25 m/s
Camera FPS	30 Hz
Max support resolution	1280 × 720
Communication interface	Modbus TCP

## Data Availability

The data that support the findings of this study are available from the corresponding author, upon reasonable request.

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
