# Peer review of "A Machine Vision Method for Identifying Blade Tip Clearance in Wind Turbines"

_sensors, 2024, doi:10.3390/s24185935_

Round 1
Reviewer 1 Report
Comments and Suggestions for Authors
The authors present a machine vision-based method for measuring the blade tip clearance in a wind turbine. In my opinion, the method is novel. There are several issues to be settled:
1. "Machine vision" would be more appropriate than "Machine vision-based method" in the keywords.
2. In the Introduction, the authors said "it is clear that there is a need for a cost-effective system for measuring the blade tip clearance, which can operate in all weather conditions, requires minimal daily maintenance, and gives clearance in real time, while mitigating the problem of low transmission delay to the controller of the wind turbine." Whether the machine vision-based system satisfies the requirements? It is suggested related indexs should be listed in the experiments.
3. In the experiments, the authors measured the clearances according to the proposed method. It is suggested the true value of the clearances and the measured errors of the proposed method should be given.
Comments on the Quality of English Language
In some conditions, passive voice is more appropriate. For example, "OpenCV is employed to develop a blade dynamic recognition algorithm in Section 4", rather than "Section 4 employs OpenCV to develop a blade dynamic recognition algorithm".
Reviewer 2 Report
Comments and Suggestions for Authors
In this manuscript, the authors summarized and discussed a turbine blade tip clearance measurement method based on machine vision. This work provides new insights into the development of wind turbine measurement. I propose the following review comments:
1. Is there a clear standard for the clearance between the blade and the tower body ?
2. Discuss the advantages of machine vision method for blade clearance measurement compared with other methods.
3. A more explicit explanation of the clearance calculation diagram will enable readers to understand the experimental process more clearly.
4. Whether experimental verification can be carried out several times to reduce experimental data errors?
Comments on the Quality of English Language
Minor editing of English language required.
Round 2
Reviewer 1 Report
Comments and Suggestions for Authors
The authors have modified the manuscript according to the reviewer's comments. But there are some text editing errors. The authors should check carefully.
Author Response
Comments: The authors have modified the manuscript according to the reviewer's comments. But there are some text editing errors. The authors should check carefully.
Respond: The author has proofread the paper carefully. Some minor syntax errors have been corrected. The missing serial number in the paper is added. Thank you very much for taking time to review this manuscript. Please find the detailed responses in the re-submitted files.
Reviewer 2 Report
Comments and Suggestions for Authors
I have no more new comments.
Comments on the Quality of English Language
I have no more new comments.
Author Response
Comment: I have no more new comments.
Responds: Thank you very much for taking time to review this manuscript.